🔓 | Applied and Industrial Microbiology | Research Article

# Reconstructing the transcriptional regulatory network of probiotic *L. reuteri* is enabled by transcriptomics and machine learning

Jonathan Josephs-Spaulding,[1] Akanksha Rajput,[2] Ying Hefner,[2] Richard Szubin,[2] Archana Balasubramanian,[2] Gaoyuan Li,[2] Daniel C. Zielinski,[2] Leonie Jahn,[1] Morten Sommer,[1] Patrick Phaneuf,[1] Bernhard O. Palsson[1,2]

**ABSTRACT**  *Limosilactobacillus reuteri*, a probiotic microbe instrumental to human health and sustainable food production, adapts to diverse environmental shifts via dynamic gene expression. We applied the independent component analysis (ICA) to 117 RNA-seq data sets to decode its transcriptional regulatory network (TRN), identifying 35 distinct signals that modulate specific gene sets. Our findings indicate that the ICA provides a qualitative advancement and captures nuanced relationships within gene clusters that other methods may miss. This study uncovers the fundamental properties of *L. reuteri*'s TRN and deepens our understanding of its arginine metabolism and the co-regulation of riboflavin metabolism and fatty acid conversion. It also sheds light on conditions that regulate genes within a specific biosynthetic gene cluster and allows for the speculation of the potential role of isoprenoid biosynthesis in *L. reuteri*'s adaptive response to environmental changes. By integrating transcriptomics and machine learning, we provide a system-level understanding of *L. reuteri*'s response mechanism to environmental fluctuations, thus setting the stage for modeling the probiotic transcriptome for applications in microbial food production.

**IMPORTANCE**  We have studied *Limosilactobacillus reuteri*, a beneficial probiotic microbe that plays a significant role in our health and production of sustainable foods, a type of foods that are nutritionally dense and healthier and have low-carbon emissions compared to traditional foods. Similar to how humans adapt their lifestyles to different environments, this microbe adjusts its behavior by modulating the expression of genes. We applied machine learning to analyze large-scale data sets on how these genes behave across diverse conditions. From this, we identified 35 unique patterns demonstrating how *L. reuteri* adjusts its genes based on 50 unique environmental conditions (such as various sugars, salts, microbial cocultures, human milk, and fruit juice). This research helps us understand better how *L. reuteri* functions, especially in processes like breaking down certain nutrients and adapting to stressful changes. More importantly, with our findings, we become closer to using this knowledge to improve how we produce more sustainable and healthier foods with the help of microbes.

**KEYWORDS**  *L. reuteri*, probiotic, machine learning, transcriptome, systems biology

Address correspondence to Bernhard O. Palsson, palsson@ucsd.edu, or Patrick Phaneuf, phaneuf@biosustain.dtu.dk.

The authors declare no conflict of interest.

See the funding table on p. 19.

*L*imosilactobacillus reuteri (*L. reuteri*) has become a focal point in probiotic research due to its versatile roles in human health and various commercial applications. Recognizing that reuterin is an intermediate in glycerol metabolism, its function extends beyond fermentation byproduct to a potent antimicrobial and its broader efficacy in food preservation (1). Reuterin's active form, 3-hydroxypropionaldehyde, has been validated for antimicrobial activity in an array of food matrices, combating pathogens

like *E. coli* O157:H7, *S. Enteritidis*, and *L. monocytogenes* (2). Such capabilities underscore its utility in safeguarding diverse food items, thereby contributing to the food industry. *L. reuteri* has been identified as a potential probiotic strain capable of synthesizing bioactive compounds that could fortify foods with essential nutrients. Notably, it can transform linoleic acid into conjugated linoleic acid (CLA), a fatty acid with a wide array of health benefits, including anti-cancer and anti-inflammatory properties (3). Moreover, *L. reuteri* is among the Lactobacillaceae family with the capability to enhance the nutritional profile of foods by producing vitamin B2 (riboflavin), an essential vitamin that plays a crucial role in cellular metabolism (4). Harnessing these capabilities, *L. reuteri* could be utilized in the development of functional foods enriched with CLA and riboflavin, thus providing these essential nutrients directly through the diet without the need for additional supplementation. These attributes position *L. reuteri* for developing nutrient-enriched functional foods, reducing the need for dietary supplements. While *L. reuteri* is recognized for its probiotic efficacy, it is not a native component of the human-associated microbiome. However, its presence in the human gut is associated with the suppression of pathogenic microbes and the support of beneficial commensals (2, 5). *L. reuteri* has been isolated from the human gut, vagina, and breast milk, providing various health benefits pertaining to host-associated microbiome modulation (6). Specifically, the benefits of *L. reuteri* range from improving vaginosis symptoms (7), reducing serum fat/cholesterol levels (8, 9), alleviating chronic intestinal inflammation (10), ameliorating symptoms of cystic fibrosis (11), protecting the gut microbiome of extremely preterm infants (12), and even alleviating maternal depression (13), among a host of other advantages (6). However, the underlying mechanisms of action remain largely unexplored for a significant proportion of these observed benefits attributed to *L. reuteri*. Generally, the health impact of *L. reuteri* has been intricately linked to metabolic activity, which is, in turn, influenced by environmental conditions. To enrich potential insights of how *L. reuteri* adapts its gene expression in response to environmental changes, we applied the machine learning algorithm independent component analysis (ICA) to a comprehensive data set of *L. reuteri* gene expression. However, to move from a theoretical model to a confirmed understanding, we recognize the necessity to validate the predictions of our analysis through experimental verification to solidify the role of ICA and data analytics in elucidating these adaptive mechanisms. Machine learning, particularly ICA, can integrate a wide array of data inputs by extracting focal points of interest (14). In this study, we applied ICA to elucidate the activity of iModulons (15), representing groups of genes that provide a comprehensive view of broad biological functions concerning varying experimental conditions. By contributing to future regulatory network-building endeavors that aim to enhance the predictive accuracy of regulatory actions, improving our understanding of species-specific transcriptional regulatory networks (TRNs) can provide insights into molecular sensing and feedback regulatory systems (16). Conversely, reverse engineering the TRN can illuminate the organism's adaptive response to a dynamic environment (17–19). Reconstruction *of L. reuteri*'s TRN is essential for hypothesizing its adaptive mechanisms in diverse environments. While ICA aids in mapping the TRN, it is important to note that the predictive power of such computational tools remains hypothetical until experimentally validated. The current predictions provide a foundation that must be empirically tested to ascertain their accuracy and reliability in depicting regulatory interactions.

Despite the widespread utilization of *L. reuteri* for health and commercial benefits, a notable gap exists in studies attempting to elucidate the metabolic mechanisms underpinning this organism's response to dynamic conditions. As of this writing, while investigating NCBI's SRA database, it was found that only 36 studies explored *L. reuteri* gene expression in response to changing environmental conditions. Of these, only six leveraged next-generation RNA sequencing, and just one utilized RNA-seq to study *L. reuteri in vitro*. To bridge this gap, we have created the *Lactobacillus* Precision RNA-seq Expression Compendium for Independent Signal Extraction or LactoPRECISE compendium, based on a large RNA-seq data set, spanning a wide array of media conditions,

to recapitulate some of the key genome-scale features of *L. reuteri*'s TRN (Fig. 1). This compendium, comprising 117 individual experiments, enables us to interrogate gene expression regulators and characterize their activity. Unlike earlier co-expression networks such as weighted correlation network analysis, which uses standard hierarchical clustering or correlation-based approaches, our study employs a modern machine learning-based approach for clustering. This enables a more refined delineation of co-regulated gene clusters.

In this study, we successfully applied ICA to the LactoPRECISE compendium to define 35 unique iModulons, which have been used in this study to improve our understanding of *L. reuteri* transcriptional regulation. iModulon activity, inferred from each respective iModulon across various conditions, indicates the biological regulation and signaling that underlies actual activity. Therefore, these activity levels offer valuable insights into the biological function, potentially guiding strain design strategies. Expressly, we have provided insight into regulating *L. reuteri* modules related to microbial food production.

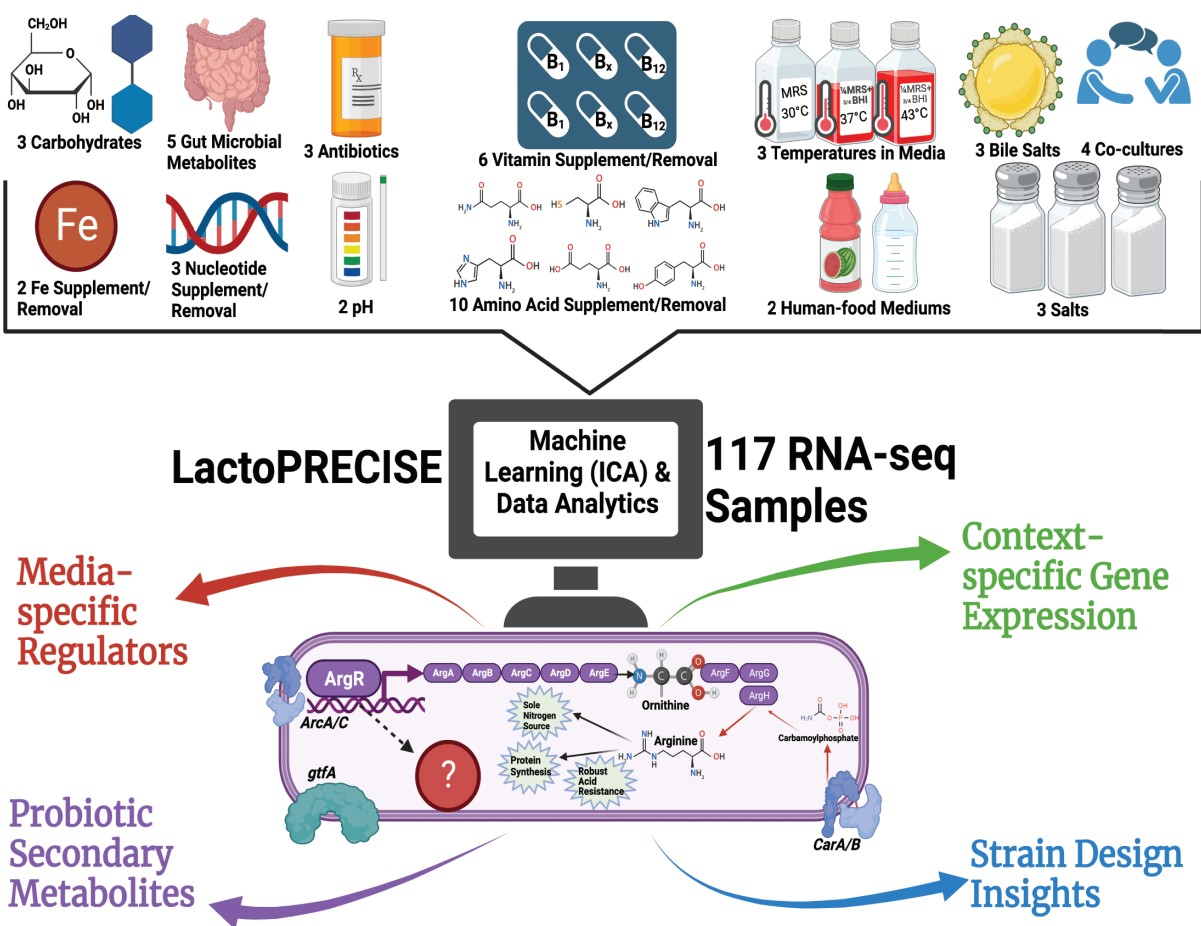

**FIG 1** Comprehensive iModulon workflow overview. Our workflow in analyzing the LactoPRECISE compendium, a curated data set containing 117 internally sequenced RNA-seq samples derived from a diversity of 50 unique conditions, encompasses an extensive range of 13 distinct condition types. We employ the machine learning algorithm independent component analysis (ICA) to discern the underlying structure of iModulons from our in-house-sequenced data set. Through data analytics, we scrutinize Modulons to uncover media-specific regulatory mechanisms governing metabolism, investigate context-dependent gene expression, and predict pathways leading to the biosynthesis of probiotic secondary metabolites. Our workflow offers a fresh view of probiotic strain design while highlighting data analytics approaches in the field.

## MATERIALS AND METHODS

### RNA extraction and library preparation

This study used *Limosilactobacillus reuteri MM4-1A* (*PTA-6475*) as the model strain. The strain was cultured in different growth media, including the base medium Chemically Defined Media (CDM), De Man-Rogosa-Sharpe (MRS), and specific food-like media, such as human milk, infant formula, or melon fruit juice, with appropriate supplements to extract a broad range of gene expression responses. We conducted experiments under aerobic or anaerobic conditions, covering variations in carbohydrate supplementation, gut microbial metabolites, salt stress, vitamin supplementation or removal, various media compositions, temperature changes, bile salt stress, co-cultures with other microbes, iron supplementation or removal, nucleotide supplementation or removal, pH changes, amino acid supplementation or removal, human food media, and antibiotic treatments. A detailed list of all the growth conditions and the exact supplementary in the respective media can be found in Table S1.

To prepare the samples, overnight cultures of the strains were grown at defined temperatures with mixing in the respective media. The cultures were then diluted to a starting optical density at 600 nm (OD600) of approximately 0.01 and incubated at defined temperatures with stirring. When the cultures reached an OD600 of 0.4, 2 mL was transferred to centrifuge tubes containing 4 mL of RNAprotect Bacteria Reagent (Qiagen). After vortexing for 5 seconds, the tubes were incubated at room temperature for 5 minutes. The samples were centrifuged for 10 minutes at $5,000 \times g$, and the supernatant was removed. Cell pellets were stored at −80℃ until further processing. In conditions involving antibiotic treatment, the antibiotics were added at 2× or 5× their minimum inhibitory concentration when the bacterial culture reached an OD600 of approximately 0.2. The cultures were then incubated at defined temperatures with stirring for an additional hour before sample collection.

Total RNA was isolated and purified from the frozen cell pellets using the Zymo Research Quick-RNA Fungal/Bacterial Microprep Kit, following the manufacturer's protocols. To remove ribosomal RNA (rRNA), 1 µg of total RNA was treated with thermostable Rnase H (Hybridase) and short DNA oligos that specifically targeted and degraded rRNA at 65℃ to preserve mRNA integrity. The resulting rRNA-subtracted RNA was converted into libraries using the KAPA RNA HyperPrep kit, which incorporated short Y-adapters and barcoded PCR primers. The libraries were quantified using a fluorescent assay and assessed for size distribution using a TapeStation. After pooling the libraries and performing a 1× SPRI bead cleanup to remove residual PCR primers, the final library pool was quantified and sequenced on an Illumina instrument (NextSeq, Novaseq).

### RNA-Seq data acquisition, processing, and formation of LactoPRECISE

Given the scarcity of *in vitro* studies on *L. reuteri* next-generation RNA-seq in public repositories, the data for LactoPRECISE were exclusively sourced from internal sequencing. All 163 raw paired-end RNA-seq samples generated for this study were subjected to the prokaryotic RNA-seq processing pipeline outlined by Sastry et al. and mapped to our reference genome (GCF_020785475.1) (20). Post-processing, the gene expression compendium was log transformed to transcripts per million (logTPM). A standardized iModulon quality control protocol is utilized; we then manually curated the experimental metadata to distinguish biological replicate samples and annotate individual samples with details such as media descriptions, treatment specifics, environmental changes, growth stages, and other pertinent growth-related parameters.

Data failing any of the four FASTQC measures: per base sequence quality, per sequence quality scores, per base n content, and adapter content were excluded from further analysis. Samples with less than 500,000 reads mapped to coding sequences were also discarded. We employed hierarchical clustering to identify samples with atypical expression profiles. After these initial quality control steps, we carried out manual metadata curation on the remaining data. We extracted details such as strain

description, base media, carbon source, treatments, and temperature from existing literature. We assigned a unique name to each project, and within each project, we used unique identifiers to recognize replicates. Post-curation, we removed samples if they (i) lacked metadata, (ii) were without replicates, or (iii) showed a Pearson $R$ correlation between replicates lower than 0.80. To determine if samples had a low correlation between replicates, gene expression data were processed through a correlation-based filtering and averaging filter to address potential dissimilarities within sample groups and their respective replicates. Specifically, gene expression data were first grouped by specific conditions. The internal consistency of each group was gauged by determining the pairwise Pearson correlation coefficient for samples within the group. To accommodate variations in gene expression data correlations, we set defined correlation thresholds at 0.8, 0.85, 0.9, and 0.95, excluding any lower thresholds due to concerns about replicate quality. If the correlation coefficient for any sample replicates within a group fell below these thresholds, indicating considerable internal variance, we averaged all samples within that group to produce a single representative expression profile. This approach yielded a filtered expression matrix, with groups with high internal dissimilarity represented by one averaged profile for further analyses. This procedure was reiterated for each correlation threshold to compare outcomes at various stringency levels of the filtering process. Notably, a threshold of $R = 0.80$ was chosen for subsequent ICA analyses. The logTPM data within each project were then centered on a reference condition specific to that project. Refer to Fig. S1A for a detailed depiction of the procedure's outcomes, illustrating the number of conditions averaged due to within-group variance.

Simultaneously, a preliminary TRN was constructed to capture a comprehensive representation of genes and their known associated regulatory elements. This process entailed accumulating known transcription factor data from RegPrecise and collating regulatory interactions from existing *L. reuteri* literature (21). The resources utilized for this draft TRN, including transcription factors, are cataloged in Table S2.

## Implementation of independent component analysis

Following the PyModulon workflow, we employed ICA on the RNA-seq compendium using the optICA extension derived from the FastICA algorithm (14). The optICA algorithm generates two matrices, M and A, necessary for determining iModulon activity. The M-matrix comprises robust components constituting the iModulons, while the A-matrix contains the activities of the iModulon. These matrices are approximations derived from the X-Matrix, representing the expression data drawn from the RNA-seq compendium.

Determining the optimal number of iModulons for LactoPRECISE requires the identification of an optimal dimensionality to circumvent both under-decomposition and over-decomposition of the gene expression compendium. To identify this optimal dimension (22), we performed multiple rounds of gene expression profile clustering within a range of 30–100, using a step size of 10. The optimal dimension of 60 was eventually selected, corresponding to the number of non-single gene independent components matching the final component count (Fig. S1B). The execution of the ICA algorithm with gene expression clustering ultimately yielded 35 robust iModulons for *L. reuteri*.

## iModulon regulon enrichment, functional annotation, and characterization

To uncover the biological roles of iModulons within LactoPRECISE, we utilized the PyModulon Python package to enrich each iModulon with known regulators identified in our preliminary TRN (20). We aligned each iModulon to our preliminary TRN table to identify iModulons with a substantial intersection with previously characterized transcriptional regulators. This step was essential to establish potential regulatory roles of identified iModulons. We assessed the relation of predicted iModulons to previously defined regulators using "iModulon recall" and "regulon recall" metrics. The "iModulon

recall" metric measures the ratio of shared genes to the total genes in an iModulon, while the "regulon recall" quantifies the ratio of shared genes to the total genes in a regulon (Fig. 2E).

Following this, to further interpret the functional context of the iModulons and their associated genes, KEGG (23) and Cluster of Orthologous Groups (COG) data were retrieved via the EggNOG mapper (24). UniProt IDs were obtained using the Uniprot ID mapper (25), operon information was gathered from Biocyc (26), and Gene Ontology

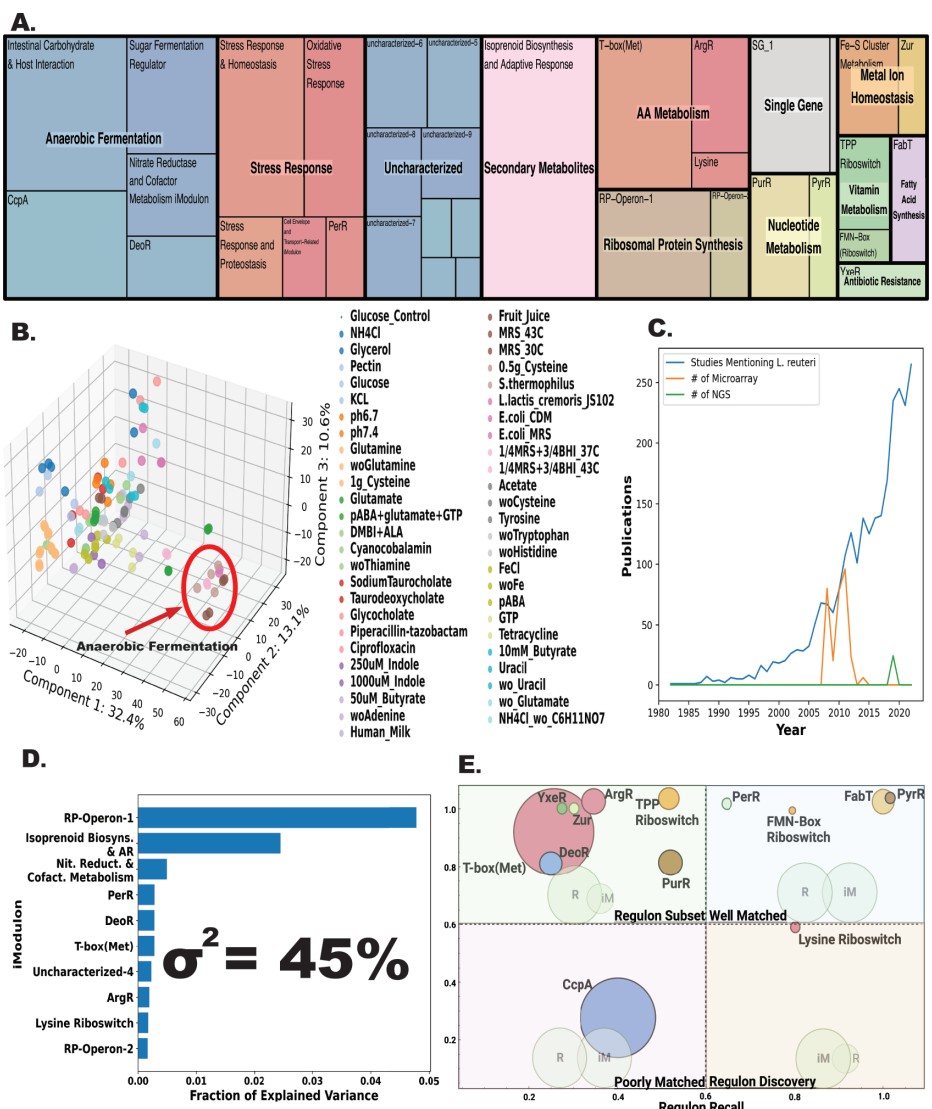

FIG 2 Description of the LactoPRECISE transcriptomics compendium. (A) A tree map depiction of the 35 identified iModulons and their associated functional categories. The dimensions of the individual boxes correlate with the number of genes found in each respective iModulon. (B) A principal component analysis plot based on the LactoPRECISE compendium gene expression highlights conditions that are similar or different. A red circle highlights conditions that undergo anaerobic fermentation. (C) Graphical representation of the volume of published research that includes the mention of "*L. reuteri*" in the abstract, plotted against the body of published microarray or next-gen RNA-seq literature. (D) The top 10 iModulons contribute to the most significant proportion of explained variance; the explained variance is described by a value of $\sigma^2$. These iModulons capture key functions such as ribosome protein synthesis, secondary metabolite biosynthesis, stress response, amino acid synthesis, fatty acid conversion, and unknown functions. (E) A scatterplot of all regulatory iModulons. Each point signifies an individual iModulon, with the size of the point proportionate to the gene count within the corresponding iModulon. Regulatory iModulons are segmented into four quadrants: regulon subset (top left), well matched (top right), poorly matched (bottom left), and regulon discovery (bottom right).

(GO) annotations were gathered from AmiGO2 (27, 28). With all gene expression data and annotations, ChatGPT 3.5 (August 3 version) was utilized in the initial characterization of uncharacterized iModulons to redefine some as functional iModulons, thereby contributing to a more comprehensive exploration of gene regulatory networks (29). Specifically, for all genes for each uncharacterized iModulon, their gene names and weight per iModulon within the compendium, potential interactions with known regulons, and COG annotations if available were input into ChatGPT (more detailed methods and conversations with ChatGPT are available in the Supplementary Methods section). ChatGPT output predictions of uncharacterized iModulon function. These predictions were verified by assessing individual gene membership, biological function, and relation to known regulons via existing publications and database to prevent integration of false-positive or "hallucination" Modulons.

## Reconstructing *L. Reuteri* ArgR bitome

Considering the lack of comprehensive information about gene-operon relations, we undertook an algorithmic approach with Bitomics to segregate genes into operons based on spatial proximity (30). This led to the creation of a data frame with a novel distance algorithm that systematically arranges *L. reuteri* genes into operon clusters. Initially, genes were batched based on their respective accession IDs. Sorted sets of genes were parsed through an iterative operation, which checked for spatial congruity with each preceding gene. If the spatial gap fell within a predefined threshold, it was assigned to the current operon; otherwise, a new operon cluster was initiated. This process ultimately led to operon clusters that can be described as lists of genes grouped based on their operon classification. This organization allowed for enhanced comprehension of the spatial arrangement and operational interactions among the genes in preparation for downstream analysis.

To form a position-specific scoring matrix (PSSM) for ArgR, we aligned the binding sites of *E. coli*'s known ArgR iModulon. We subsequently quantified the frequency of each nucleotide at individual positions, normalizing this against the background frequency in the entire sequence, which resulted in the computation of log-odds scores. The "bitome_fasta.motif_search" function was employed for the iterative computation of motif scores for equidistant segments, equivalent to the motif length. The segment with the most substantial score was earmarked as the motif score of that sequence.

Motif scores became the basis for a logistic regression model in our study, enhanced by applying elastic net regularization. We leveraged the elastic net penalty, chosen to counteract multicollinearity among predictors while fostering a lean feature selection process. This streamlined selection simplifies the interpretation of how genes interplay with ArgR. A resampling technique was also deployed to address the challenge of class imbalance, a potential pitfall in predictive modeling. Using the motif scores as the input matrix and binarized ArgR regulon memberships as target labels, a logistic regression model was built with an elastic net penalty (with an L1 to L2 ratio of 0.5) for regularization. To tackle class imbalance, we employed a SMOTETomek resampling technique, set to a parameter of k_neighbors = 5, to generate a balanced training set. To ensure the representation of each class, we utilized a fivefold cross-validation with stratified sampling. The train-test-split was conducted within the operon space, ensuring genes from the same operon were kept together. The model's performance was evaluated using an area under the receiver operating characteristic (AUC-ROC) curve, with a pre-decided threshold of 0.8 as the benchmark for satisfactory performance.

## Prediction of biosynthetic gene clusters

Biosynthetic gene clusters (BGCs) in *L. reuteri* were identified using the antiSMASH 7.0 algorithm (31) applied to the *L. reuteri* reference genome (GCF_020785475.1). We intentionally selected a "relaxed" detection strictness setting to increase the likelihood of capturing a broader array of potential BGCs since *L. reuteri* is not known to have many BGCs. AntiSMASH identified two main types of *L. reuteri* BGCs using this setting:

ribosomally synthesized and post-translationally modified peptides (RiPP) and type III polyketide synthase (T3PKS). This method also offers gene ontology annotations for the constituents of the predicted BGCs. It is worth noting that although RiPP genes were present, they appeared inactive in our compendium, leading us to focus our analysis on T3PKS.

## RESULTS

### Elucidation of the *L. reuteri* transcriptional regulatory network through transcriptomic compilation and independent component analysis

To understand the global transcriptional regulation that governs the physiological processes across a broad range of conditions, which can lead to diverse functional responses in *L. reuteri*, we applied ICA to a large RNA-seq compendium. LactoPRECISE contains 117 RNA-seq samples across a range of 50 different conditions from 13 different media types to allow for profiling that extracts diverse regulatory interactions from expression data, helping to distinguish between specific transcriptional regulators. Our analysis identified 35 integral components or iModulons, which account for 45% of the variance in gene expression across our compendium and thus represent the variation derived from transcriptional regulation (Fig. 2D).

The variance explained by the 35 iModulons is notably lower than that by principal component analysis (PCA) on the X-matrix alone (45% in ICA and 56.1% in PCA) (Fig. 2). The "anaerobic fermentation" functional group includes iModulons which likely underlie the regulatory processes of fermentation in *L. reuteri*; specifically, "nitrate reductase and cofactor metabolism," "intestinal carbohydrate and host interaction," "sugar fermentation regulator," *CcpA*, and *DeoR* comprise of 23% of our transcriptional profile. Analyzing the outlier group "anaerobic fermentation" through PCA, we found a 26.48% variance contribution from outliers in anaerobic conditions. The integration of PCA with ICA offers a nuanced view of gene regulation and function, enhancing our understanding of the data in the LactoPRECISE database. Unlike traditional coexpression methods such as weighted correlation network analysis, which are dependent on linear correlations (32) such as Pearson's correlation, ICA exposes independent sources of expression variation, capturing complex, and nonlinear relationships among genes (15). This flexibility yields a granular and adaptable method for revealing intricate regulatory networks across conditions.

Manual characterization enabled interpretations of iModulons derived from independent components as sources of biologically relevant observations in the compendium, capturing a broad spectrum of biological functional categories (Fig. 1A). Fig. S2 illustrates the correlation of iModulons across the compendium by utilizing hierarchical clustering. Fig. S3 further identifies the top five most significant iModulon clusters and highlights iModulons with the most similar cellular responses in *L. reuteri*. Notably, iModulons with known biological functions can explain the highest fraction of expression variance in our compendium including those controlling ribosomal protein synthesis (RP-Operon-1 and RP-Operon-2), secondary metabolite production (isoprenoid biosynthesis and adaptive response), nitrate reductase (nitrate reductase and cofactor metabolism), stress response (PerR), amino acid synthesis [T-box (methionine), ArgR, lysine riboswitch], a uncharacterized iModulon (uncharacterized-4), and deoxyribonucleoside synthesis operon transcriptional regulator (DeoR) (Fig. 2D).

In our initial characterization of these 35 iModulons, we found that 13 identified iModulons overlapped with known regulators, suggesting that these "regulatory iModulons" reflect the basic structure of previously known regulons. Details on each iModulon and their respective functions, number of genes, and other relevant characteristics can be found in Table S3. We also observed two additional iModulons, dominated by a single, high-coefficient gene, which could be termed "single-gene iModulons," possibly regulated solely by that gene.

We further quantified the relationship between regulatory iModulons and known regulons used to reconstruct our draft TRN using metrics such as "iModulon recall"

and "regulon recall" (Materials and methods). Most identified regulatory iModulons show a high iModulon recall rate, indicating strong agreement between the draft TRN structure and the newly identified iModulons for *L. reuteri* (Fig. 2E). High iModulon recall and regulation recall values denote a consistency between a particular iModulon and a previously characterized regulon. The overlap between the gene lists in computed iModulons and regulons within each quadrant is visualized by background Venn diagrams in each quadrant. Regulon recall is the ratio of shared genes between an iModulon and a regulon to the total genes in a regulon. In contrast, iModulon recall is the proportion of shared genes to the overall genes in an iModulon.

Next, we attempted to characterize the remaining 20 "uncharacterized iModulons" using data from public databases such as KEGG, COG, UniProt, Biocyc, and GO annotations. These iModulons were defined through various approaches, such as investigating gene membership, functional gene annotations, iModulon activity against various conditions, and ChatGPT 3.5. While ICA exhibits a high success rate in identifying biologically valuable modules of the TRN, many remain uncharacterized; however, utilization of ChatGPT improved our ability to describe uncharacterized iModulons functionally. Despite concerns about ChatGPT's potential for delivering inconsistent information in the form of "hallucinations," the language model proved valuable in synthesizing overlooked gene annotations which may have required higher levels of expert insights. This enhanced the curation of uncharacterized iModulons and led to the characterization of 11 functional iModulons (stress response and homeostasis, cell envelope and transport-related, RP-Operon-1, nitrate reductase and cofactor metabolism, sugar fermentation regulator, intestinal carbohydrate and host interaction, Fe-S cluster metabolism, RP-Operon-2, oxidative stress response, isoprenoid biosynthesis and adaptive response, stress response and proteostasis); these functional iModulons contain gene groups related to specific biological functions, but not directly linked to a known transcriptional regulator. Finally, the remaining nine uncharacterized iModulons that could not be adequately characterized can be ideal targets for future research; specifically, the iModulon "uncharacterized-9" will be presented later in the manuscript as involved in the regulation of nitrogen metabolism (see Fig. 7). This analysis has led to the systematic characterization of at least 26 iModulons. It has provided a broad range of functional descriptions that can be used to compare iModulons to their respective gene clusters.

## Visualizing *L. reuteri*'s iModulons: deciphering key regulatory roles in the probiotic landscape

*L. reuteri*'s relatively small number of iModulons is likely due to its compact genome (33); we have provided a visualization to display each iModulon's specific role in regulating this probiotic organism. Fig. 3 presents the top 10 iModulons that contribute to the greatest explained variance in the LactoPRECISE compendium to display the relative influence and condition-specific behavior of each iModulon. Each subplot displays both an iModulon's gene membership, a histogram that plots gene expression and frequency within that iModulon, and if those genes are regulated by a known regulon. Next to each histogram, a color-coded bar plot signifies the respective iModulon's activity levels against different condition types. These iModulons contribute to the highest explained variance and possibly the most crucial functions in the LactoPRECISE compendium: RP-Operon-1, isoprenoid biosynthesis and adaptive response, nitrate reductase and cofactor metabolism, PerR, DeoR, T-box (methionine), uncharacterized-4, ArgR, lysine riboswitch, and RP-Operon-2. For a complete view of the remaining 25 iModulons with histograms and activity plots within the LactoPRECISE compendium, refer to the Supplementary Figures (Fig. S4).

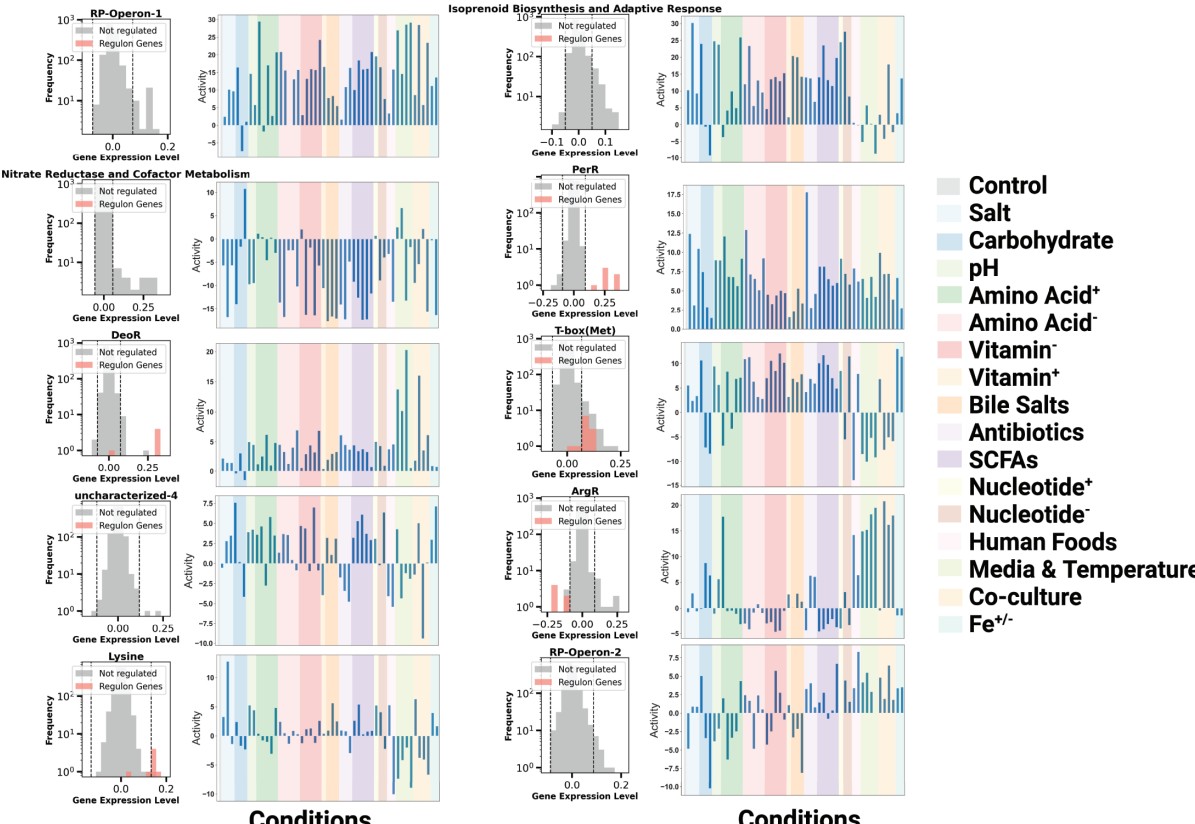

**FIG 3** Visualization of the top 10 iModulons with the highest contribution to the explained variance in the LactoPRECISE compendium. Each subplot presents an iModulon with a histogram detailing gene frequencies across various expression levels followed by the respective iModulon activity level across all conditions. This visual representation provides insights into the relative influence and condition-specific behavior of each iModulon.

## Bistable controls between fatty acid conversion and flavin mononucleotide production uncovered by iModulon analysis

Understanding control mechanisms over intricate metabolic networks is crucial to progressing microbial strain design for desired probiotic metabolites. Regulatory networks modulate the production of essential compounds like vitamins, fatty acids, and natural products in the desired quantities (34–38). While several other iModulons may contribute more toward explained variance in the LactoPRECISE compendium, the additional value lies in improving the knowledge of iModulons useful in the probiotic industry. Microbial fatty acid conversion can present unique challenges, such as rate-limiting steps or specific micronutrient requirements (39, 40). FMN, a precursor to riboflavin (vitamin B2), plays a vital role in human health by supporting metabolic processes, and its synthesis in probiotics contributes to improved gut health and nutrient absorption (41). Riboflavin and fatty acid conversion may be likely regulated by both the FMN-Riboswitch iModulons and the FabT iModulon, which both potentially play a pivotal role in influencing a two-step synthesis process for these desirable products in microbial foods. Besides nutritional fortification advantages, riboflavin might also influence the regulation and conversion of various fatty acids (42–46).

The analysis of Fig. 4A and B reveals co-regulation of FabT and FMN-Riboswitch iModulons in *L. reuteri*, modulated by specific media components. Thiamine limitation or *E. coli* coculture suppresses both iModulons, while glucose or human milk elevates their activities (47–51). Despite *L. reuteri's* known limitations in starch metabolism, as highlighted by its reliance on reuteransucrases for synthesizing reuteran from sucrose and glucanotransferase for converting starch into soluble polymers without the production of usable oligosaccharides, our observed responses suggest

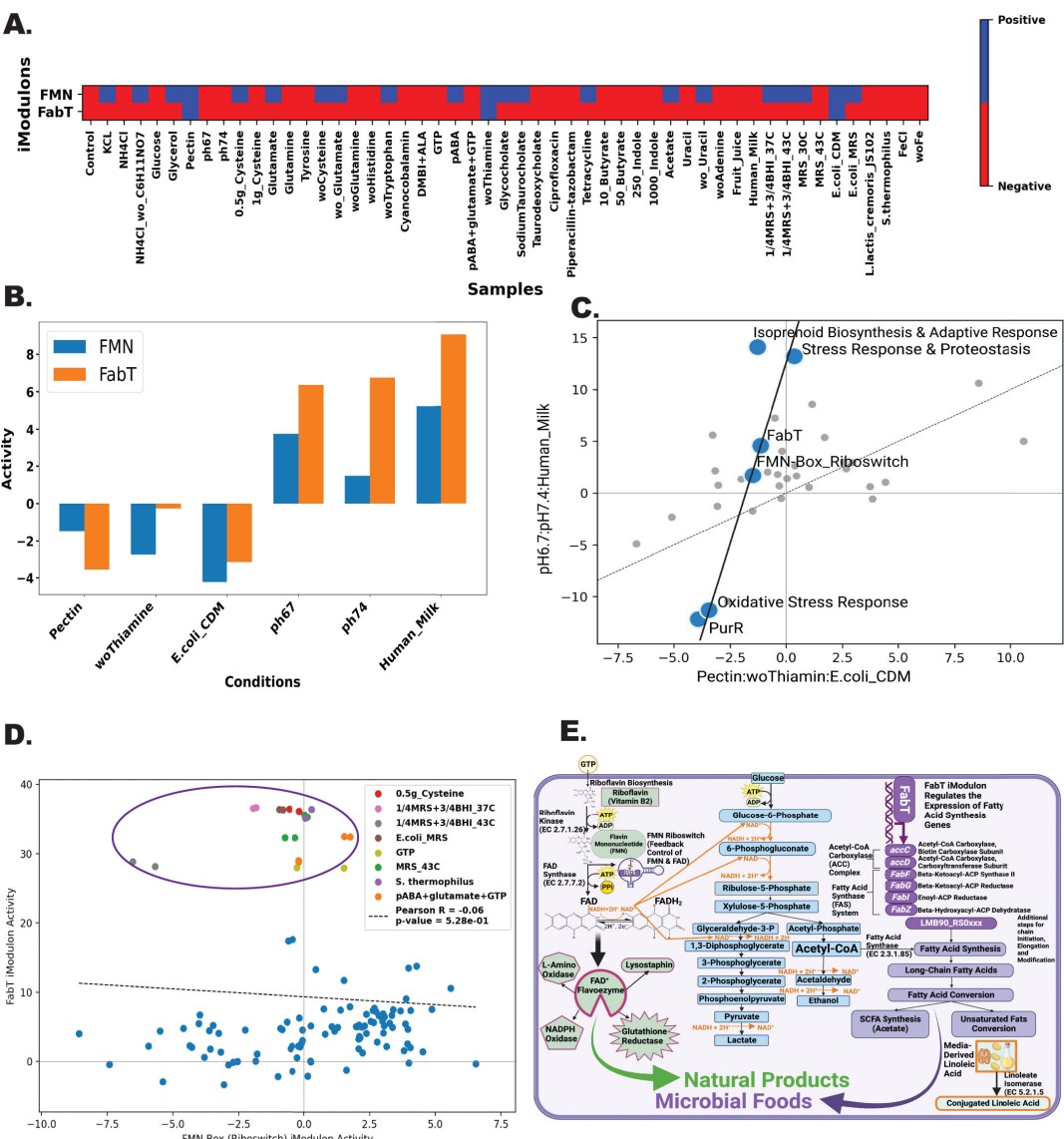

**FIG 4** Bistable regulation and interactive metabolic pathways between FabT and FMN-Riboswitch iModulons in producing microbial foods and natural products. (A) Binary heatmap showing the two distinct activities of the FabT and FMN-Riboswitch iModulons across various conditions. The color gradient illustrates the relative activity level of the iModulons, with red indicating positive activity and blue indicating negative. Overlapping conditions likely indicate a transition state between the two iModulons. (B) A bar plot represents conditions exhibiting similar activity between the FMN-Riboswitch and FabT iModulons. The conditions are represented on the x-axis, and the corresponding iModulon activity level is on the y-axis. This similarity highlights the potential bistability of regulation under these specific conditions. (C) Differential iModulon activity (DIMA) plot depicting the interactions and shared regulatory trends between various iModulons, including those for stress response and purine production. The distinctive trend line connecting FabT and FMN-Riboswitch iModulons reflects their potential bistable regulation. (D) Scatter plot illustrating the iModulon activity of FabT plotted against FMN Riboswitch. The distinctive cluster at the top suggests a particular set of conditions where both iModulons are simultaneously active, providing evidence for a potential "third state" in the bistable regulation. (E) Schematic diagram depicting a model of interconnected metabolic pathways involved in producing microbial foods (mediated by FabT) and natural products (mediated by FMN Riboswitch). This figure highlights the importance of these iModulons in the overall metabolic network and their potential roles in bistable regulation.

broader carbohydrate metabolic capabilities. This potentially involves indirect regulatory mechanism influences by complex carbohydrate interactions. Such findings underscore a nuanced regulatory interplay, pivotal for understanding *L. reuteri*'s metabolic functions and optimizing strain design.

This observation is further strengthened by a differential iModulon activity plot that illustrates peculiar but similar trajectories for FabT and FMN-Riboswitch iModulon under the conditions which regulate both FabT and FMN-Riboswitch activities (Fig. 4C). The most interesting is the grouping of two iModulons associated with stress response mechanisms (oxidative stress response and stress response and proteostasis) and natural product production during stressful conditions (isoprenoid biosynthesis and adaptive response) which may highlight the role of stress response in the production of microbial foods and natural production. Moreover, Fig. 4D and A scatter plot illustrating the iModulon activity of FabT plotted against FMN Riboswitch reveals a distinctive cluster of conditions at the top (purple outlining). This cluster suggests a particular set of conditions where both iModulons are simultaneously active. This provides evidence for a potential "third state" in the bistable regulation, likely due to specific regulatory conditions. Specifically, it should be noted that these highlighted conditions that activate this state are either media supplemented with riboflavin precursors or anaerobic fermentation conditions.

Fig. 4E proposes a model of bistable regulation between the FMN-Riboswitch and FabT iModulons. The crosstalk from these iModulons potentially forms a feedback loop driven by riboflavin supplementation, which enhances fatty acid conversion of linoleic acid into CLA in probiotics (52). Delving deeper, riboflavin is already established to modulate the kinetic stability of lipids by acting as a antioxidant (53). Interestingly, in non-prokaryote models, there is a documented influence of vitamins, either through restrictions or supplements, on fatty acid conversion (54, 55). In the context of our study, this could suggest that the FMN Riboswitch, upon activation, promotes FAD production. This production might initiate various metabolic events such as energy production, fatty acid breakdown, and flavoenzyme generation. Overall, this hypothetical model portrays iModulons in action that may play an important role in a more complicated metabolic regulatory system, which may benefit microbial food and natural product synthesis. However, there is a lack of substantial experimental results that support this interpretation of riboflavin and fatty acid interplay. While our model presents a promising blueprint, rigorous experimental probing is vital to refine, validate, or challenge our present understanding.

## Fine-tuning arginine production: exploring the role of medium components on *L. reuteri* strain design

The activity of the ArgR iModulon under various conditions was investigated by examining the expression of selected genes that are well annotated in *L. reuteri* (Fig. 5). Given the propensity of *L. reuteri*, an arginine auxotroph, to convert arginine to ornithine and considering the likely broader implications of the ArgR iModulon in acid resistance and biofilm formation, this iModulon presents as a particularly compelling case study for our investigation and merits focused attention in future research (56). By comparing individual gene expression against iModulon activity, we identified conditions that act as activators or repressors of the ArgR iModulon. These observations were further strengthened by using Pearson $R$ correlation across the genes *arcA* (Pearson $R = 0.89$, $P$-value $<1^{-10}$), *arcC* (Pearson $R = 0.89$, $P$-value $<1^{-10}$), *argF* (Pearson $R = 0.89$, $P$-value $<1^{-10}$), *argH* (Pearson $R = -0.94$, $P$-value $<1^{-10}$), *carA* (Pearson $R = -0.83$, $P$-value $<1^{-10}$), and *gtfA* (Pearson $R = 0.13$, $P$-value $<1.48^{-1}$), thus providing strong correlation into the conditions activating or repressing the ArgR iModulon with high confidence (except for *gtfA*). Conditions like co-culturing with other lactic acid bacteria (LAB) strains (*L. lactic cremoris JS102* or *S. thermophilus*) or supplementing fruit juice consistently activated the ArgR iModulon. Interestingly, in our study, we observed that while iron is not essential for the growth of most *Lactobacillus* strains, its absence in the growth medium notably influences the expression of *argF/H* and, subsequently, the activity of the ArgR iModulon. This finding suggests the existence of a regulatory mechanism in *Lactobacillus* that responds to iron availability, impacting specific gene expressions and modulating metabolic pathways. Overall, the limitation of various nucleotides or amino

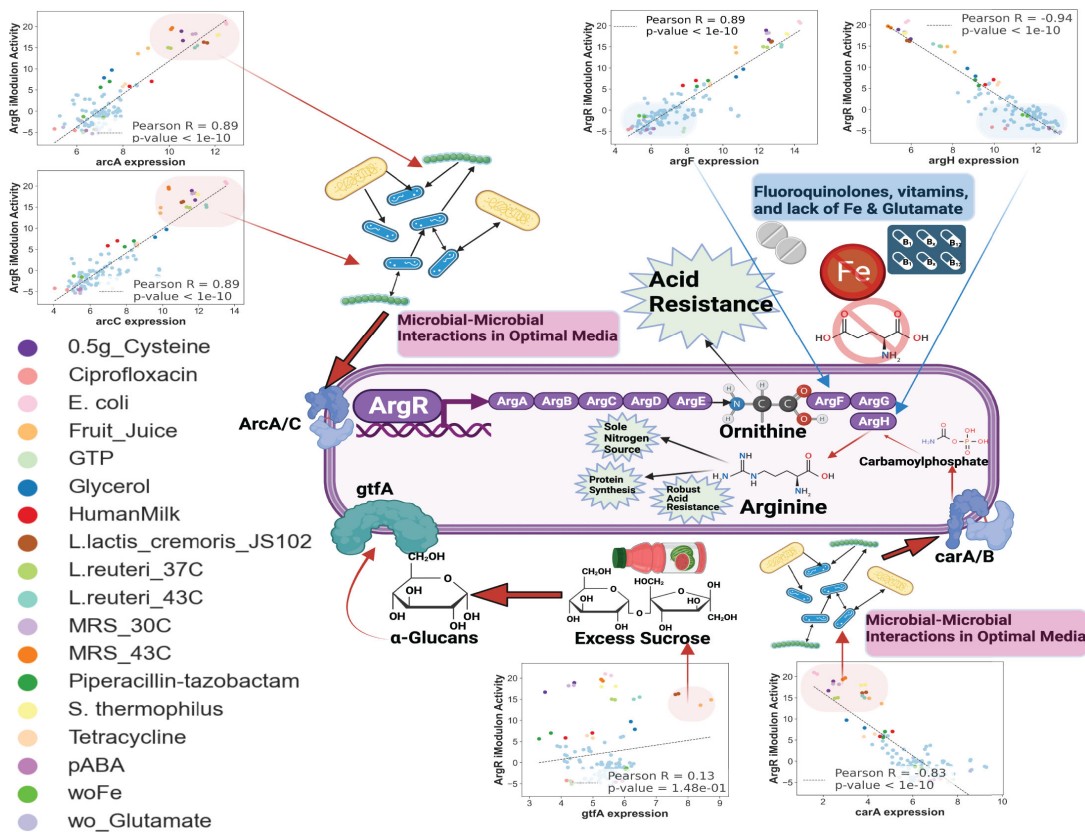

**FIG 5** Insight into the context-specific metabolic networks of the ArgR iModulon. This figure presents an illustrative diagram of the ArgR iModulon, a key iModulon in the LactoPRECISE compendium. This iModulon comprises various genes, including *arcA*, *arcC*, *argF*, *argH*, *carA*, and *gtfA*. Alongside the schematic, individual gene expression plots corresponding to the activity levels of the ArgR iModulon are also displayed. We observed positive correlations between the activity of the ArgR iModulon and context-specific gene expression of ArgR gene members such as *arcA*, *arcC*, *argF*, and *gtfA*, suggesting that these genes are co-activated when arginine synthesis is heightened. In contrast, genes *argH* and *carA* display a negative correlation, with high expression when ArgR iModulon activity is low, indicating their potential role in regulating the arginine synthesis process.

acids, exposure to antibiotics, and iron limitation were found to repress the activity of the *ArgR* iModulon.

## Bitomics analysis illuminates ArgR regulatory complexity in *L. reuteri*

To enhance our previous analysis focusing on the *L. reuteri* ArgR iModulon and media constraints, a Bitomics methodology that utilized machine learning and comparative structural motifs was used to further decipher insights into the dynamic regulatory mechanisms governed by the ArgR iModulon (30). This analytical process deepened our comprehension of the regulatory mechanisms underlying *L. reuteri*'s ArgR iModulon. By employing gene weight plotting to compare the *E. coli* and *L. reuteri* ArgR iModulons, differences were found between the respective gene clusters (Fig. 6). While *E. coli* ArgR is primarily related to "amino acid transport and metabolism," the *L. reuteri* ArgR gene membership displayed a broader spectrum of COG functions (carbohydrate transport and metabolism, energy production and conversion, inorganic ion transport and metabolism, transcription, translation/ribosomal structure and biogenesis, and genes with unknown functions or lacking COG annotations) and negative gene weights across the genes in the ArgR iModulon. These variations highlight the possibility of a more intricate regulatory landscape within *L. reuteri* ArgR marked by diverse functions and negative gene weights despite the shared ArgR motif with *E. coli* (Fig. 6A and B).

A position-specific scoring matrix (PSSM) was also generated to identify the distribution of nucleotide frequencies at each position within the ArgR binding sites.

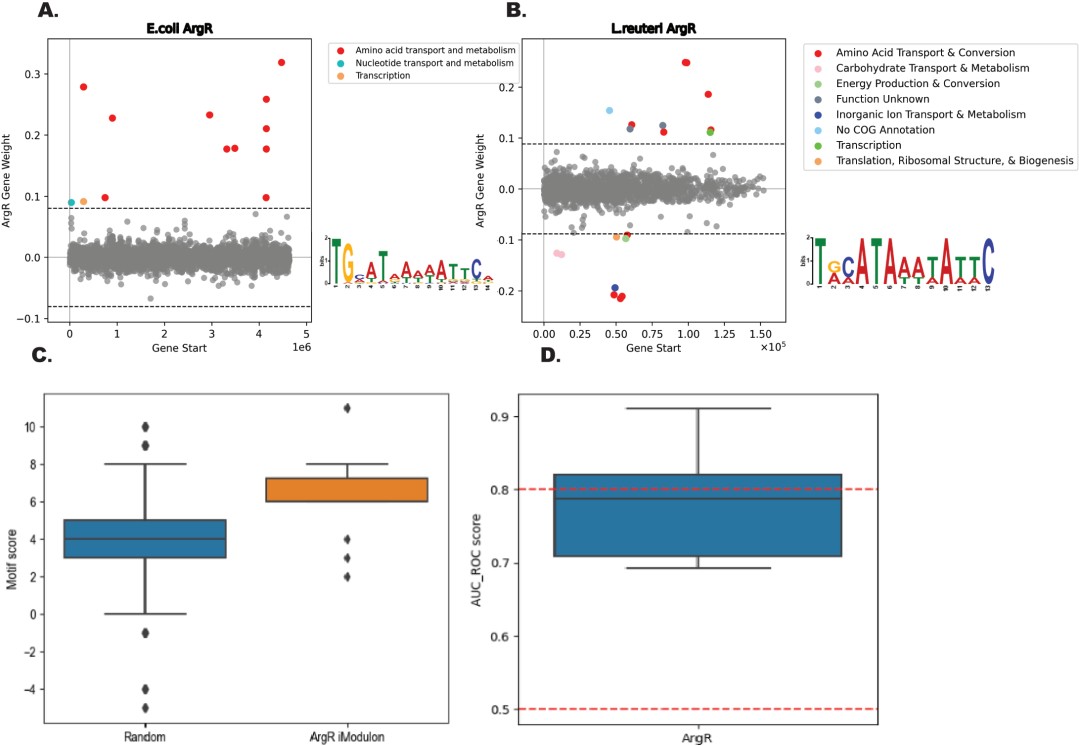

**FIG 6** Decoding ArgR regulation in *L. reuteri* through structural motif and Bitomics techniques. (A) Gene weight plot for *E. coli* ArgR iModulon. In *E. coli* ArgR, most genes pertain to "amino acid transport and metabolism." (B) Gene weight plot for *L. reuteri* ArgR iModulon. In *L. reuteri* ArgR, a broader diversity in Clusters of Orthologous Groups (COG) functions and negative gene weights is observed. This suggests a more complex regulatory scenario despite the similar ArgR motif between the two organisms. (C) Analysis of *ArgR* motif strength in *ArgR* regulated genes versus random genes. This plot compares ArgR motif prevalence and strength among genes regulated by *ArgR* and a random selection of genes. Using a PSSM distinguished nucleotide distributions in ArgR binding sites, revealing genes within the ArgR motif had markedly higher motif scores than those outside; this distinction was further accentuated when comparing ArgR-regulated genes to a random gene cohort. (D) Area under the receiver operating characteristic (AUC-ROC) score assesses the performance of this study's computational model. A high AUC-ROC score reflects the model's capacity to classify genes correctly under the ArgR regulation based on motif strength. The combined analysis presented in this figure illustrates the complexity of *ArgR* regulation in *L. reuteri* and points toward potential differences in regulatory strategies compared to *E. coli*, a thoroughly studied model organism.

We identified significant motifs within the gene sequences using the log-odds scores derived from the PSSM. A marked distinction was observed in the motif scores of genes residing within and outside the ArgR motif. The enhanced motif scores within the ArgR iModulon of *L. reuteri* confirm the precision of our model in capturing the *ArgR* regulatory network, which is underscored by the identification of genes like arginosuccinate synthase (*LMB90_RS03525*), argininosuccinate lyase (*argH*), and arginine deiminase (*arcA*). These genes, with their high motif scores, are critical for arginine conversion and catabolism, highlighting arginine's central role in *L. reuteri*'s metabolic processes. The presence of the arginine repressor gene (*LMB90_RS04695*) within this iModulon further illustrates the stringent control over these pathways, which are integral to the organism's energy production and adaptation to stress, demonstrating the multifaceted influence of *ArgR* in *L. reuteri*'s physiology.

A comparison of ArgR motif prevalence and strength between genes regulated by ArgR and a random gene selection further underscored this finding (Fig. 6C). The model's effectiveness is validated by an AUC-ROC score approaching 0.8 (Fig. 6D), a supportive metric in the model's ability to predict ArgR regulatory patterns. Selecting AUC-ROC as our performance metric provides a panoramic view of the model's discrimination capabilities, which spans various classification thresholds to ensure that our assessment remains balanced.

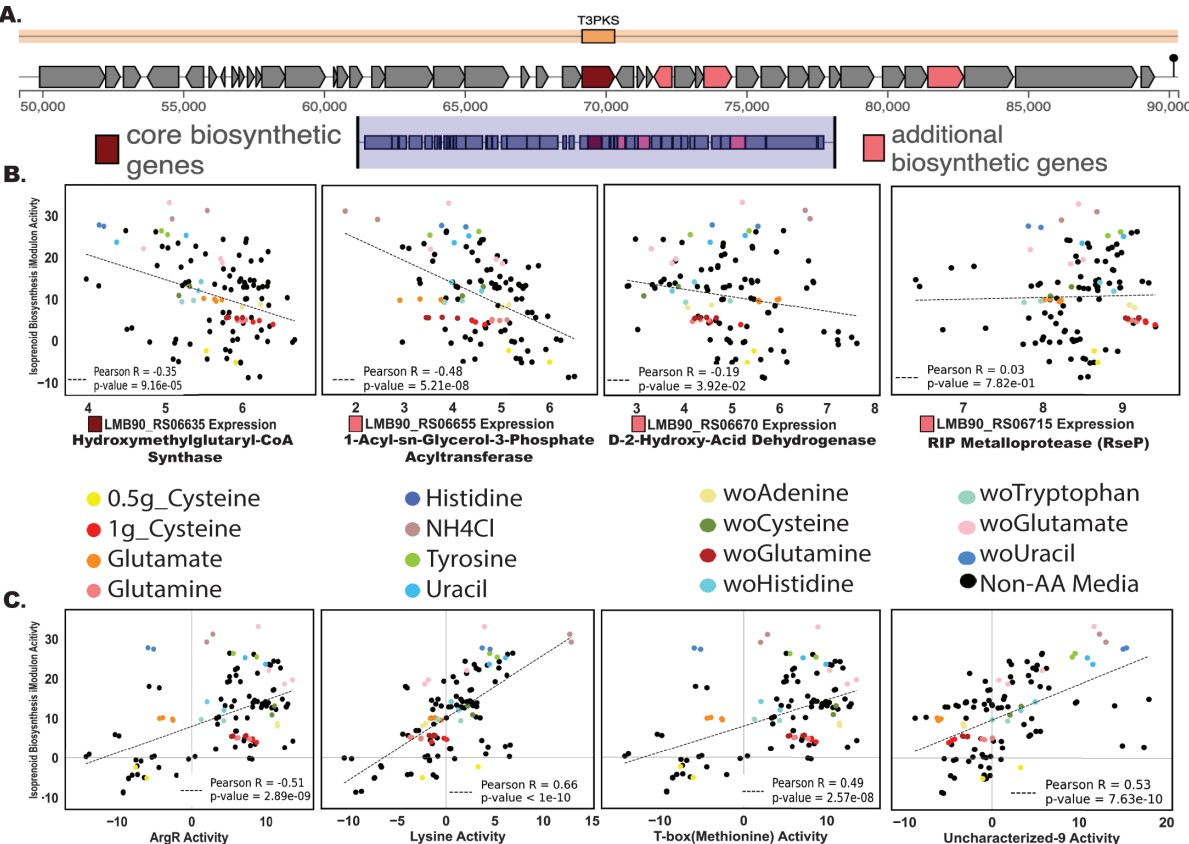

**FIG 7** Unraveling the interplay between the type III polyketide synthase (T3PKS) gene cluster and iModulon activity in *L. reuteri,* with an emphasis on nitrogen metabolism and isoprenoid biosynthesis. (A) Displays the predicted biosynthetic gene clusters (BGCs) from the *L. reuteri* genome, as predicted by antiSMASH 7.0. The focus is on the T3PKS BGC, which comprises one core biosynthetic gene and three additional biosynthetic genes that are active in our compendium. (B) We present the gene expression of these four biosynthetic genes plotted against the "isoprenoid biosynthesis and adaptive response" iModulon. The plots underscore the strong activation of this iModulon under conditions related to nitrogen metabolisms, such as removing glutamate or uracil and supplementing with NH4Cl. (C) Further exploration of this iModulon activity to three well-characterized iModulons [*ArgR*, lysine riboswitch, and T-box (methionine)], all of which are known to regulate amino acid metabolism, as well as the "uncharacterized-9" iModulon, encompassing genes involved in ammonia transport. Strong correlations emerge between the "isoprenoid biosynthesis and adaptive response" iModulon and these four iModulons, reinforcing the role of nitrogen metabolism in regulating secondary metabolite production. This figure illuminates potential strategies for engineering *L. reuteri* to optimize the production of secondary metabolites by modulating conditions that highly activate key iModulons.

## Linking nitrogen metabolism to secondary metabolite production in *L. reuteri*

We investigated nitrogen metabolism and the potential to regulate isoprenoid biosynthesis by observing the interplay between the T3PKS BGC and iModulon activity within *L. reuteri*. T3PKS enzymes synthesize polyketides, a large family of secondary metabolites with diverse biological activities and therapeutic potential. This iModulon comprises genes vital for isoprenoid production, which are essential components of cell membranes and play important roles in an organism's stress response by producing secondary metabolites (57). Our examination of the T3PKS gene cluster, illustrated in Fig. 7A and identified by antiSMASH 7.0 (31), spotlights the four active genes responsible for polyketide synthesis. Specifically, the core biosynthetic gene (*LMB90_RS06655*) codes for hydroxymethylglutaryl-CoA synthase, while the three additional biosynthetic genes (*LMB90_RS06655*, *LMB90_RS06670*, and *LMB90_RS06715*) code for 1-acyl-sn-glycerol-3-phosphate acyltransferase, D-2-hydroxy-acid dehydrogenase (also found in the sugar fermentation regulator iModulon), and RIP metalloprotease *RseP*, respectively. This cluster represents a group of secondary metabolites known for their biological diversity and potential therapeutic utility (57).

Interestingly, the "isoprenoid biosynthesis and adaptive response" iModulon demonstrated diverse activity when plotting with the expression of T3PKS's four biosynthetic genes across nitrogenous conditions (Fig. 7B). Under specific conditions tied to nitrogen metabolism, including glutamate or uracil depletion and NH4Cl supplementation, this iModulon was notably active. Fig. 7C provides a deeper association into the intricate relationships between the isoprenoid biosynthesis and adaptive response iModulon and four other distinct iModulons: ArgR, lysine riboswitch, and T-box (methionine) stand out as established regulators of amino acid metabolism. Concurrently, the "uncharacterized-9" iModulon, containing numerous genes tied to ammonia transport, will likely have similar functions as the other amino acids regulating iModulons. Hierarchical clustering of LactoPRECISE's most common iModulon clusters brings further insight (Fig. S3); cluster 2 (Spearman's $R = 0.20$) encompasses the iModulons "isoprenoid biosynthesis and adaptive response," "lysine riboswitch," and "uncharacterized-9." This alignment hints at a synchronized regulation and a shared response across the gene expression compendium. Our identification of the "isoprenoid biosynthesis and adaptive response" functional iModulon in *L. reuteri* indicates a predicted involvement in isoprenoid pathways, yet it does not confirm the active biosynthesis of isoprenoids or carotenoids. This finding presents a potential biological role that necessitates further research for confirmation and understanding, highlighting the exploratory nature of iModulon analysis in revealing novel microbial functions.

## DISCUSSION

Our comprehensive analysis of *L. reuteri*'s TRN illuminated the identification of 35 distinct iModulons, offering deeper insights into the molecular mechanics underpinning its functions. Among these, 13 were discerned as regulatory iModulons with ties to well-established transcription factors (TFs). This finding not only augments our current understanding of TF-gene dynamics but also bridges present knowledge between potential co-regulators and regulatory targets. Their complicated roles in stress response, along with amino acid and secondary metabolite production, underscore their importance in tailoring effective probiotics. This study's significant finding is the identification of 11 unique functional iModulons from 20 uncharacterized ones in *L. reuteri*. iModulons enriched with gene clusters for specific biological roles that are not yet linked to known transcriptional regulators offer a promising area for future research. Their coordinated response to environmental stimuli underscores their potential in revealing novel regulatory mechanisms and advancing microbial strain design. The 11 functional iModulons identified in LactoPRECISE and their descriptions are presented in the Supplementary Discussion.

Specifically, our data analytic approach upon our large iModulon data set has identified several metabolic pathways which have been shown to be either non-functional or are unlikely to exist in *L. reuteri*. Most importantly, these iModulons were identified to be "functional iModulons," a group of iModulons which have not been yet identified to be regulated by a transcriptional regulator, but rather a biological response to diverse conditions. For example, the "nitrate reductase and cofactor metabolism" functional iModulon contains various genes within the NAR operon (*narI*, *narJ*, and *narH*) and the molybdenum cofactor-dependent *mobB*; even though most *L. reuteri* strains lack nitrate respiration, these nitrate reductase genes hint toward the possibility of conditional expressions. Further studies are required to better understand their functions across various nitrogen conditions. Our study also observed a "Fe-S cluster metabolism" functional iModulon; iron assimilation is atypical for *L. reuteri* and the *Lactobacillus* as a taxonomic group. However, our study identified genes contained within the *Suf* operon (*sufB*, *sufC*, and *sufD*) in addition to *feoB*; these genes point to a potential metabolic function or environmental adaptation that merits further in-depth investigations to uncover if regulation of metal homeostasis does play an important physiological role in *L. reuteri*. Finally, the "isoprenoid biosynthesis and adaptive response" functional iModulon contained the genes *mvaD* and *mvk*, which are related to the potential

biosynthesis of isoprenoid precursors. However, without the supplementation of more direct *in vitro* evidence of isoprenoid biosynthesis, the likelihood of this organisms producing isoprenoids is speculative and merits further research. These three functional iModulons and the other predicted functional iModulons require further investigation and evidence to better understand their potential role of their metabolic pathways and physiological roles in *L. reuteri*.

Observing potential co-regulation between select iModulons offers insights into shared functionalities, which are essential for advanced strain design efforts. The bistable regulation observed in the FabT/FMN iModulons highlights the complexities of regulatory mechanisms that are crucial for metabolic functions (58, 59). This concept is supported by mathematical models of bistability in microbial systems, as seen in the *SigE* regulatory network in *Mycobacteria* and the *Bacillus subtilis* biofilms, where bistable emergence of oscillations and stress responses demonstrate similar regulatory dynamics (60, 61). Additionally, phenotypic bistability in *E. coli*'s central carbon metabolism further underscores the feasibility of such mechanisms in prokaryotic systems (62). In our proposed model, metabolic reconstruction was used as a foundational step, delineating biochemical reactions and pathways specific to our organism of interest. We hypothesize the interaction between three major metabolic pathways: riboflavin biosynthesis, phophoketolase pathway, and the FabT iModulon, providing a comprehensive framework for our analysis. The FMN-Riboswitch acts as a pivotal feedback control in this network, influencing the conversion of FAD into FADH2 and thus intersecting with the phophoketolase pathway through the utilization of NAD$^+$; given that *Lactobacillus* is anaerobic and does not utilize ATP through beta oxidation, this may be a promising way for energy utilization in this organism (63). This interplay is crucial as it links riboflavin metabolism directly to energy production and fatty acid regulation. The phophoketolase pathway, pivotal in our model, is likely to progress with NAD utilization from FAD conversion into FADH2, leading to lactate and ethanol production and support for fatty acid regulation—a key aspect to be validated. The intersection of this pathway with fatty acid regulation is particularly intriguing. The FabT iModulon regulates the expression of genes essential for fatty acid synthesis, such as *AccC*, *accD*, *FabF*, *FabG*, *FabI*, and *FabZ*, which interact with acetyl-CoA, highlighting the intricate network of metabolic processes (64). Furthermore, our model suggests that the biosynthesis of vitamins such as riboflavin and short-chain fatty acids, such as acetate, are not just metabolic end products but also serve as substrates for microbial-based foods (65, 66). This extends to transforming dietary linoleic acid into CLA via linoleate isomerase, indicating a broader scope of fatty acid and vitamin metabolism in microbial systems (66, 67). Further influence by dietary elements such as starch and pectin, despite *L. reuteri*'s limited direct starch metabolism and lack of pectinase, possibly hints at a broader carbohydrate metabolic capability, likely through indirect gene expression and cell wall regulation (68). Overall, our hypothetical model, supported by insights from the interconnectedness of these pathways and their regulatory elements, proposes a dynamic and complex metabolic regulatory system. This system could have significant implications for microbial food and natural product synthesis. However, the current lack of substantial experimental results that specifically support this comprehensive interpretation highlights the need for rigorous experimental investigation. Our model, while promising, requires further refinement, validation, or challenge to enhance our understanding of the regulatory response of *Lactobacillus* spp. to an expanded variety of media (69, 70).

Our analysis provides potential insights toward understanding of the *ArgR* metabolic pathway. The observed metabolism of abundant glutamate during co-cultivation with various LABs may indicate symbiotic cross-feeding between strains, as inferred from our data (71). However, to firmly establish this understanding, further experimental validation is necessary to confirm this observation's causative relationship and underlying mechanisms. Our results also revealed that *L. reuteri* possesses active glutaminases, which likely play a pivotal role in the organism's acid resistance. The deamidation of

glutamine by these enzymes not only aids in the maintenance of intracellular pH but also contributes to the overall acid tolerance of the bacterium by increasing the expression of arginine-ornitinine transporters (72). The significance of glutaminases in the metabolism of glutamine and the ensuing acid resistance is corroborated by Li et al., who highlighted the importance of these enzymes in *L. reuteri* and other lactobacilli adapted to vertebrate hosts, suggesting a conserved mechanism among these bacteria (73). Furthermore, the catabolism of arginine alongside glutamine is recognized as a general strategy for acid resistance in LAB, including *L. reuteri*, reflecting a broader adaptive response to acidic stress (74). The interplay between these metabolic pathways emphasizes the complex regulatory networks that underlie the acid resistance phenotype in *L. reuteri* and potentially in other related probiotic strains. The GH70 family of glycoside hydrolases, which encompasses the *gtfA*-encoded reuteransucrase, is integral to *L. reuteri*'s ability to process sucrose, into α-glucans. This enzymatic activity is crucial for the synthesis of extracellular polysaccharides that facilitate biofilm formation and host adhesion (75–77). In conjunction with this, it has been demonstrated that glycerol presence upregulates the *carA* and *carB* genes in *L. reuteri*, pivotal in the synthesis of carbamoyl-phosphate. While this pathway traditionally leads to arginine synthesis, this may not be a primary route in *L. reuteri*. Nevertheless, the interplay between *gtfA*-mediated sucrose conversion to α-glucans and the *carA/B*-driven carbamoyl-phosphate pathway highlights a complex metabolic network. This network is not only integral to enhancing *L. reuteri*'s acid resistance but also contributes to supplying essential amino acids to the host. Moreover, this sophisticated metabolic interplay can potentially improve the shelf-life and freshness of fermented foods, like bread, by optimizing sucrose utilization and fostering the production of reuteran-like polymers in *L. reuteri* (77–79).

Our Bitomics analysis, enhanced with machine learning and comparative structural motifs, has revealed a more intricate landscape of ArgR regulation in *L. reuteri* than previously understood, especially when contrasted with *E. coli*. While *E. coli*'s ArgR primarily relates to amino acid transport and metabolism, *L. reuteri*'s ArgR demonstrates a broader range of functions. This divergence highlights the unique regulatory pathways of *L. reuteri* in arginine metabolism, distinct from those in *E. coli*. The high AUC-ROC score reinforces the model's predictive strength. These findings, emphasizing the complex interplay between ArgR iModulon activity and gene expression under various conditions, offer valuable insights for future strain design and applications.

Exploration of nitrogen metabolism and isoprenoid biosynthesis in *L. reuteri*, utilizing AntiSMASH analysis, has uncovered a nuanced interplay between the T3PKS BGC and various iModulon activities. The isoprenoid biosynthesis and adaptive response iModulon, potentially engaged in synthesizing crucial isoprenoids like ubiquinones and carotenoids, displayed varied activity under different nitrogen conditions, reflecting a sophisticated regulatory network. The hierarchical clustering analysis revealed coordinated regulation among multiple iModulons, indicative of a strategic metabolic equilibrium responsive to nitrogen availability. Importantly, the "additional biosynthetic genes" identified in the T3PKS BGC, such as *LMB90_RS06655* (involved in lipid biosynthesis), *LMB90_RS06670* (part of the sugar fermentation regulator iModulon), and *LMB90_RS06715* (potentially involved in polyketide synthesis signaling pathways), elucidate the intricate contributions to the metabolic versatility of *L. reuteri*. These findings point toward *L. reuteri*'s metabolic adaptability and underscore the potential of metabolic engineering in enhancing isoprenoid and polyketide production for pharmaceutical and nutritional applications. Additionally, the potential of nitrogenous media to activate select iModulons hints at enhancing secondary metabolite production, aligning with industrially relevant bioprocesses, thereby emphasizing the real-world applicability of our insights. The diverse correlations between the "isoprenoid biosynthesis and adaptive response" iModulon and its four iModulon counterparts across numerous nitrogen media underscore an intricate narrative that is speculative in nature. While our identification of this iModulon suggests a predicted involvement in isoprenoid pathways, it does not conclusively establish the active biosynthesis of isoprenoids or

carotenoids; this finding, therefore, underscores a potential, yet unconfirmed, biological role that could be leveraged in strain design to influence isoprenoid biosynthesis in *L. reuteri*. The exploratory nature of iModulon analysis, as revealed in this study, highlights the need for further research to fully understand and potentially harness these complex metabolic interactions (80).

## Conclusion

With iModulons presenting a promising frontier in probiotic microbial foods and natural product engineering, the potential applications and interpretations are vast. Our research has provided an informed, data-driven approach to strain design, emphasizing the importance of continually updating and refining our knowledge base for improved reconstruction of microbial TRNs. As our understanding evolves, the aim remains to harness the vast potential of microbial physiological systems, ushering in innovative methods to redefine the future of microbial foods.

### ACKNOWLEDGMENTS

We thank Omkar Satyavan Mohite and Omid Ardalani for the informative discussion on biosynthetic gene clusters and *Lactobacillus reuteri* metabolism. We thank Marc Abrams for reviewing the manuscript and providing constructive suggestions. We thank Kevin Rychel for providing support throughout the project on iModulon curation and analysis. We are also thankful to Rebecca Höög for improving the quality of all figures. Fig. 3E, Fig. 4, and Fig. 7 were created with BioRender.

### AUTHOR AFFILIATIONS

[1]The Novo Nordisk Foundation Center for Biosustainability, Technical University of Denmark, Copenhagen, Denmark
[2]Department of Bioengineering, University of California, San Diego, California, USA

### AUTHOR ORCIDs

Jonathan Josephs-Spaulding http://orcid.org/0000-0001-9828-0685
Morten Sommer http://orcid.org/0000-0002-8308-948X
Patrick Phaneuf http://orcid.org/0000-0002-4122-6589
Bernhard O. Palsson http://orcid.org/0000-0003-2357-6785

### FUNDING

| Funder | Grant(s) | Author(s) |
|---|---|---|
| Novo Nordisk Fonden (NNF) | NNF20CC0035580 | Bernhard O. Palsson |

### AUTHOR CONTRIBUTIONS

Jonathan Josephs-Spaulding, Conceptualization, Data curation, Formal analysis, Investigation, Methodology, Software, Validation, Visualization, Writing – original draft, Writing – review and editing | Akanksha Rajput, Data curation, Methodology, Software, Writing – review and editing | Ying Hefner, Methodology | Richard Szubin, Methodology | Archana Balasubramanian, Formal analysis, Methodology, Software, Visualization | Gaoyuan Li, Formal analysis, Methodology, Software | Daniel C. Zielinski, Data curation, Formal analysis, Investigation, Methodology, Software | Leonie Jahn, Conceptualization, Methodology, Resources, Writing – review and editing | Morten Sommer, Conceptualization, Funding acquisition, Resources | Patrick Phaneuf, Methodology, Project administration, Resources, Software, Supervision, Writing – original draft, Writing – review and editing | Bernhard O. Palsson, Conceptualization, Funding acquisition, Project administration, Resources, Supervision, Writing – original draft, Writing – review and editing

## DATA AVAILABILITY

All in-house-generated sequences were deposited in the NCBI-Sequence Read Archive database (PRJNA989027). The accession number of the deposited reads is provided in Table S1. The prokaryotic RNA-seq processing pipeline is available at https://github.com/avastry/modulome-workflow. The optICA algorithm employed in the manuscript is available at https://github.com/avsastry/modulome-workflow/tree/main/4_optICA. The X, M, and A matrices, in addition to the TRN regulator file, gene ontology, iModulon Table Files, and code to produce these, are all available on GitHub (https://github.com/PoeticPremium6/LactoPRECISE). A DOI with the code used in this manuscript can also be found on Zendo (https://zenodo.org/record/8108422). Each gene and iModulon have interactive, searchable dashboards on iModulonDB.org (https://imodulondb.org/dataset.html?organism=l_reuteri&dataset=lactoprecise), and data can also be downloaded from there.

## ADDITIONAL FILES

The following material is available online.

### Supplemental Material

**Supplemental Figures (mSystems01257-23-S0001.docx).** Figures S1 to S4.
**Supplemental Methods (mSystems01257-23-S0002.pdf).** Utilization of ChatGPT for iModulon characterization.
**Supplemental Discussion (mSystems01257-23-S0003.docx).** Discussion of the 10 identified functional iModulons.
**Supplemental Tables (mSystems01257-23-S0004.xlsx).** Tables S1 to S3.

### Open Peer Review

**PEER REVIEW HISTORY (review-history.pdf).** An accounting of the reviewer comments and feedback.

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
