## [Reviewer comments · mSystems]

Reconstructing the Transcriptional Regulatory Network of Probiotic *L. reuteri* is Enabled by Transcriptomics and Machine Learning

Jonathan Josepchs-Spaulding, Akanksha Rajput, Ying Hefner, Richard Szubin, Archana Balasubramanian, Gaoyuan Li, Daniel Zielinski, Leonie Jahn, Morten Sommer, Patrick Phaneuf, and Bernhard Palsson

Corresponding Author(s): Patrick Phaneuf, Danmarks Tekniske Universitet The Novo Nordisk Foundation Center for Biosustainability

Review Timeline:

Submission Date:	November 27, 2023
Editorial Decision:	December 21, 2023
Revision Received:	January 4, 2024
Accepted:	January 9, 2024

Editor: Danilo Ercolini

Reviewer(s): The reviewers have opted to remain anonymous.

Transaction Report:

DOI: <https://doi.org/10.1128/msystems.01257-23>

Re: mSystems01257-23 (**Reconstructing the Transcriptional Regulatory Network of Probiotic *L. reuteri* is Enabled by Transcriptomics and Machine Learning**)

Dear Dr. Patrick Phaneuf:

Revision Guidelines

Sincerely,
Danilo Ercolini
Editor
mSystems

Reviewer #2 (Comments for the Author):

The manuscript presents an innovative bioinformatic platform to identify genome-wide transcriptional regulatory networks in *Lm. reuteri*. Analyses are based on a very large RNA-sequencing dataset that were generated with a broad range of environmental conditions. While the manuscript has improved in the revision, the interpretation of the data should be more carefully grounded by a revised discussion of metabolic pathways that have not been shown to be functional in *Lm. reuteri* (nitrate respiration) or

are unlikely to exist (isoprenoid synthesis, synthesis of linoleic acid or SCFA other than acetate, beta oxidation). This discussion should be placed in the main manuscript, not in a "supplementary discussion" section.

Specific comments.

line 53 to 57. The references on CLA and riboflavin synthesis pertain to *Lactiplantibacillus plantarum* and *Lactococcus lactis* but not to *Lm. reuteri*.

line 101. While it is appreciated that the manuscript describes a very large dataset, quantitative comparisons that allow a sound statistical analysis are generally based on triplicate independent observations.

Line 290. Anaerobic respiration. Anaerobic respiration requires a nitrate / nitrite reductase and an electron transfer chain. While both genes are occasionally present in strains of *Lm. reuteri*, it is uncertain that both are present in the same strain and are functional. Absent at least a bioinformatic confirmation that all genes that are necessary for anaerobic respiration are present, the use of the term "anaerobic respiration" is not warranted. Nitrate reductase may alternatively serve to recycle reduced co-factors that are generated in the phosphoketolase pathway.

line 308. Isoprenoid biosynthesis in lactobacilli is well documented as pigmented lactobacilli synthesize the carotenoids through isoprenoid precursors. The genes in *Lm. reuteri* is less than 30% homologous to known isoprenoid biosynthetic genes in lactobacilli, or to any other biochemically characterized isoprenoid biosynthetic enzyme. The prediction of the role of the gene in isoprenoid biosynthesis is merely speculation.

line 371. *Lm. reuteri* converts fatty acids but is not known to synthesize fatty acids.

line 382 / 383. *Lm. reuteri* is not known to produce short chain fatty acids other than acetate and certainly does not synthesize linoleic acid.

line 389. Unclear. Carbohydrate metabolism in *Lm. reuteri* is very well characterized. Malto-oligosaccharides are utilized but starch is not as the only extracellular glycosyl hydrolases are reuteransucrases, which use sucrose to synthesize reuteran while releasing fructose, and glucoamylases, which convert starch to a soluble polymer but not to oligosaccharides which could be used as carbon source.

line 409. Beta-oxidation of fatty acids for ATP generation has not been described for *Lm. reuteri*. As many strains of the species do not tolerate oxygen very well, beta-oxidation is an unlikely metabolic pathway in this organism.

line 482. This reviewer remains unconvinced that any strain of *Lm. reuteri* produces isoprenoids or carotenoids from isoprenoid precursors.

LINE 585. Arginine biosynthesis is an unlikely event in *Lm. reuteri*.

line 600. Isoprenoid biosynthesis, see above.

Figure 4E. *Lm. reuteri* does not produce SCFA other than acetate. Saturated long chain fatty acids are not converted to linoleic acid.

Response to Reviewers

January 4, 2024

Title: Reconstructing the Transcriptional Regulatory Network of Probiotic *L. reuteri* is Enabled by Transcriptomics and Machine Learning

Manuscript ID: mSystems01257-23

JF Josephs-Spaulding, A Rajput, Y Hefner, R Szubin, A Balasubramanian, G Li, D Zielinski, LJ Jahn, MOA Sommer, PV Phaneuf, and BO Palsson

Thank you for taking the time to provide these critical and insightful comments, which have significantly bolstered the quality of our resubmitted manuscript. We have thoroughly responded to each comment that the peer reviewers provided and used these comments to improve the quality of our manuscript for future readership. For simplified reading, the comments provided by the peer reviewers will remain black, and our responses to each remark will use **blue** text color. Additionally, any manuscript text that was edited to address the peer reviewer's concerns will be displayed in a **red** text color in our newly uploaded '**Marked-Up Manuscript**.'

Reviewer 2:

- The manuscript presents an innovative bioinformatic platform to identify genome-wide transcriptional regulatory networks in *Lm. reuteri*. Analyses are based on a very large RNA-sequencing dataset that were generated with a broad range of environmental conditions. While the manuscript has improved in the revision, the interpretation of the data should be more carefully grounded by a revised discussion of metabolic pathways that have not been shown to be functional in *Lm. reuteri* (nitrate respiration) or are unlikely to exist (isoprenoid synthesis, synthesis of linoleic acid or SCFA other than acetate, beta oxidation). This discussion should be placed in the main manuscript, not in a "supplementary discussion" section.
 - **The authors are thankful for your additional comments in this second round of revision and for continuing to improve the quality of this manuscript greatly. We have responded to each of your comments with care. More so, we have revised the mention of the mentioned metabolic pathways throughout the results and added a new paragraph within the main-text discussion section, which discusses 3 Functional iModulons. With regards to the specific metabolic pathways that you have mentioned, we have now regrouped the 'Nitrate Reductase and Cofactor Metabolism' iModulon within the 'Anaerobic Fermentation' functional group rather than 'Anaerobic Respiration.' These changes can now be found in Lines 297-299 and Figure 1A. We have added several statements in an attempt to prevent confusion that our 'Isoprenoid**

Biosynthesis and Adaptive Response' Functional iModulon is not in a regulator that controls the synthesis of isoprenoids or carotenoids but rather contains the genes that underlie this potential biological function; this iModulon and all 'Functional iModulons' are repeatedly clarified to have these characteristics and the need for future verification of these predictions. Numerous mentions of this, our 'Isoprenoid Biosynthesis and Adaptive Response' Functional iModulon, can be found in Lines 23, 514-515, 545-550, 581-585, and 706-712. We are hopeful that these additions to our manuscript have convinced the reviewer and future readers that we are not claiming the regulation of Isoprenoid Biosynthesis but the observation of isoprenoid-regulated genes with expression that is modulated by diverse nitrogenous conditions. We have made sure to remove any mention of CLA production or biosynthesis that was missed in the previous revision and made sure only to mention the conversion or transformation from linoleic acid to CLA (Line 611). We have made sure to describe acetate and not other SCFAs as being derived from *L. reuteri*. We have removed any inappropriate mention of beta-oxidation. This change can be found in Line 611 and Figure 3E. Discussion about these metabolic pathways and others, which have gone into greater detail in the 'Supplementary Discussion' section, have now been added to the main text. However, we would like to retain the 'Supplementary Discussion' section not only to control the Main Text word count but also to provide the readership with a compact way to read further into our predicted 'Functional modules.' The reviewer can now find Lines 566-585 to contain an expanded discussion of 3 Functional iModulons and their associated metabolic pathways (Nitrate Reductase, Iron-Sulfur Metabolism, and Isoprenoid Biosynthesis), which are unlikely to be functional or exist in *L. reuteri* to be added. We have also highlighted to the reader in this section and throughout the text that these Functional iModulons require further verification to tie their prediction to functional and physiological roles in *L. reuteri*.

- line 53 to 57. The references on CLA and riboflavin synthesis pertain to *Lactiplantibacillus plantarum* and *Lactococcus lactis* but not to *Lm. reuteri*.
 - **Thank you for providing this suggestion to update our citations to describe the bioconversion of linoleic acid to conjugated linoleic acid and riboflavin production, both by *L. reuteri*. Reference numbers 3 and 4 have now been updated.**
- line 101. While it is appreciated that the manuscript describes a very large dataset, quantitative comparisons that allow a sound statistical analysis are generally based on triplicate independent observations.

- **The authors are thankful that the reviewer has decided to continue the conversation about describing our ‘high-quality’ sequencing datasets as being based on triplicates. Therefore, we have removed the wordage for ‘high-quality’ in Lines 105 (replaced with ‘large’) and 287 (replaced with ‘large’).**
- Line 290. Anaerobic respiration. Anaerobic respiration requires a nitrate / nitrite reductase and an electron transfer chain. While both genes are occasionally present in strains of *Lm. reuteri*, it is uncertain that both are present in the same strain and are functional. Absent at least a bioinformatic confirmation that all genes that are necessary for anaerobic respiration are present, the use of the term "anaerobic respiration" is not warranted. Nitrate reductase may alternatively serve to recycle reduced co-factors that are generated in the phosphoketolase pathway.
 - **The authors appreciate Reviewer 2’s technical knowledge in supplementing our analysis, which has greatly benefitted the quality of this manuscript throughout the peer review process. Thank you for providing additional insight into our usage of ‘anaerobic respiration’. We have now modified this to ‘Anaerobic Fermentation’ as a functional group that contains multiple iModulons which likely underlie the regulatory processes of fermentation in *L. reuteri* (Lines 297-298). To comply with this modification, we have updated Figure 1A with this word usage.**
- line 308. Isoprenoid biosynthesis in lactobacilli is well documented as pigmented lactobacilli synthesize the carotenoids through isoprenoid precursors. The genes in *Lm. reuteri* is less than 30% homologous to known isoprenoid biosynthetic genes in lactobacilli, or to any other biochemically characterized isoprenoid biosynthetic enzyme. The prediction of the role of the gene in isoprenoid biosynthesis is merely speculation.
 - **Yes, the authors completely agree that the role of the genes within the ‘Isoprenoid Biosynthesis & Adaptive Response’ iModulon are all ‘Functional iModulons’ are purely speculation. We have attempted to establish this important point throughout the manuscript. All ‘Functional iModulons’ i.e. iModulons that are not directly related to a known transcription factor or regulator, are speculative to describe a potential function; the value of ‘Functional iModulons’ lies to in the future of additional studies to unravel the potential functions that we have predicted. Meanwhile, ‘Functional iModulons’ and their limitations are described in our manuscript (Lines 361-363 & 559-562). To further clarify to the reviewer and readership, we have added more information in Line 23, Lines 545-550, and Lines 695-701 to inform the readers that our present prediction of isoprenoid biosynthesis in *L. reuteri* is purely speculative.**

- line 371. *Lm. reuteri* converts fatty acids but is not known to synthesize fatty acids.
 - **The authors thank Reviewer 2 for reiterating their concerns from the previous revisions with regard to our description of 'Fatty Acid Synthesis.' We have now ensured that the wordage 'Fatty Acid Conversion' or 'Fatty Acid Regulation' is used. These changes can be found on Lines 21, 397, 444, 605, and 607**
- line 389. Unclear. Carbohydrate metabolism in *Lm. reuteri* is very well characterized. Malto-oligosaccharides are utilized but starch is not as the only extracellular glycosyl hydrolases are reuteransucrases, which use sucrose to synthesize reuteran while releasing fructose, and glucanotransferases, which convert starch to a soluble polymer but not to oligosaccharides which could be used as carbon source.
 - **Thank you for this suggestion to clarify the mechanisms underlying various carbohydrate sources and interactions. Our statement now suggests that despite *L. reuteri* containing specific enzymatic pathways (such as reuteransucrases and glucanotransferases), this organism likely exhibits broader carbohydrate metabolic pathways. Lines 405 -411 now reflect these changes.**
- line 409. Beta-oxidation of fatty acids for ATP generation has not been described for *Lm. reuteri*. As many strains of the species do not tolerate oxygen very well, beta-oxidation is an unlikely metabolic pathway in this organism.
 - **Thank you for identifying this artifact we failed to remove in our last revision. Therefore, we have adjusted this statement to reflect our citation (Lines 441-442) better.**
- line 482. This reviewer remains unconvinced that any strain of *Lm. reuteri* produces isoprenoids or carotenoids from isoprenoid precursors.
 - **We appreciate the reviewer's concerns regarding the biosynthesis of isoprenoids and carotenoids in *L. reuteri*. However, we have continuously expanded our discussion of these 'Functional modules' throughout the manuscript's main text. The 'Isoprenoid Biosynthesis and Adaptive Response' Functional iModulon identified in our study suggests a potential biological role in isoprenoid pathways, but it is not definitive evidence of active biosynthesis. This iModulon, like others identified, indicates coordinated gene clusters potentially involved in specific biological functions yet not directly linked to known transcriptional regulators. Hence, our findings highlight a predicted role, underscoring the need for further research to confirm and elucidate these pathways in *L. reuteri*. Therefore, we have now expanded numerous areas of our manuscript to clarify that our model's prediction of isoprenoid biosynthesis is speculative, not tied to transcriptional regulators, and requires further research to verify its connection to the change of isoprenoid-related genes**

across various conditions; this is a common feature of 'Functional iModulons' and adjustments can be found on Lines 23, 514, 545-550, 566-570, 581-587, and 695-701.

- LINE 585. Arginine biosynthesis is an unlikely event in *Lm. reuteri*.
 - Thank you for this suggestion; the authors agree since we have mostly attempted to describe the regulation of arginine in *L. reuteri* during this manuscript. We have now adjusted the phrasing of our paragraph to highlight that the traditional pathway for arginine synthesis may not be the most primary and there is likely a more sophisticated metabolic interplay. This has been added to Lines 657–666.
- line 600. Isoprenoid biosynthesis, see above.
 - Thank you for your concern with the 'Isoprenoid Biosynthesis and Adaptive Response' Functional iModulon. As with our previous statement regarding this iModulon, this is a 'Functional iModulon' and not a 'Regulatory iModulon'. This means that this iModulon is potentially involved in a specific biological function, but it is not linked to known regulators. There is a need for future research to explore this and verify these predictions. However, to prevent uncertainty from the readers, we have added expanded statements throughout the manuscript to highlight that this predicted module does not verify or confirm the biosynthesis of isoprenoids or carotenoids but rather contains genes involved in this pathway. Please see Lines 23, 514, 545-550, 566-570, 581-587, and 695-701 for this expanded material.
- Figure 4E. *Lm. reuteri* does not produce SCFA other than acetate. Saturated long chain fatty acids are not converted to linoleic acid.
 - Thank you for providing this comment to supplement your previous remarks on SCFAs in *L. reuteri*. We have now adjusted Figure 3E, which correctly contains mention of acetate, rather than the nonexistent Figure 4E to reflect this change in SCFA production. The authors agree that saturated LCFAs are not converted into linoleic acid and believe that the reviewer has misinterpreted the figure; this misinterpretation may have arisen due to the proximity of the CLA to 'Unsaturated Fats Conversion', however, there is no arrow connecting these two. We have attempted to make this more clear by visually exaggerating the border color of the foods that can supply 'linoleic acid' and the end-product of CLA as well. We have also specified in bold font that this is 'Media-Derived Linoleic Acid'. Thank you for your attention in improving this detail in the Figure 3E.

Re: mSystems01257-23R1 (**Reconstructing the Transcriptional Regulatory Network of Probiotic *L. reuteri* is Enabled by Transcriptomics and Machine Learning**)

Dear Dr. Patrick Phaneuf:

Your manuscript has been accepted, and I am forwarding it to the ASM production staff for publication. Your paper will first be checked to make sure all elements meet the technical requirements. ASM staff will contact you if anything needs to be revised before copyediting and production can begin. Otherwise, you will be notified when your proofs are ready to be viewed.

Featured Image Submissions: If you would like to submit a potential Featured Image, please email a file and a short legend to mSystems@asmusa.org. Please note that we can only consider images that (i) the authors created or own and (ii) have not been previously published. By submitting, you agree that the image can be used under the same terms as the published article. File requirements: square dimensions (4" x 4"), 300 dpi resolution, RGB colorspace, TIF file format.

Sincerely,
Danilo Ercolini
Editor
mSystems